# Laser-Induced Thermal Annealing of CH₃NH₃PbI₃ Perovskite Microwires

**Xiaoming Chen** [1], **Zixian Wang** [2], **Ren-Jie Wu** [3], **Horng-Long Cheng** [3,*] and **Hsiang-Chen Chui** [1,3,*]

1   School of Optoelectronic Engineering and Instrumentation Science, Dalian University of Technology, Dalian 116024, China; chen_xm@dlut.edu.cn
2   School of Microelectronics, Dalian University of Technology, Dalian 116024, China; X59654862@mail.dlut.edu.cn
3   Department of Photonics, National Cheng Kung University, Tainan 70101, Taiwan; l78041151@ncku.edu.tw
*   Correspondence: shlcheng@mail.ncku.edu.tw (H.-L.C.); hcchui@dlut.edu.cn (H.-C.C.)

**Abstract:** Perovskite microwires have a larger surface-to-volume ratio and better photoelectric conversion efficiency than perovskite films. The degree of crystallization also affects the optoelectrical performances of perovskite microwires. Laser annealing was regarded as a tool for crystallization. High light absorption induced fast heating process. A 405 nm violet laser located near the absorption peak of typical perovskite films was employed as the annealing laser. In an in situ experimental design, the annealing laser beam was combined into the micro Raman measurement system. Real-time information of the annealing and crystallization was provided. Many excellent works were done, and typically needed offline optoelectronic measurements. An mW-level continuous-wave laser beam can provide enough kinetic energy for crystalline in perovskite microwires. The thermal distribution of the perovskite microwire under the annealing laser beams was considered here. Polarized Raman signals can provide evidence of the perovskite microwires crystallization. This work offered the novel approach of an on-site, real-time laser-induced thermal annealing design for perovskite microwires. This approach can be used in other material procedures. Intensity-dependent conditions were crucial for the annealing processes and analyzed in detail. The substrate effect was found. This proposed scheme provided integrated novel, scalable, and highly effective designs of perovskite-based devices.

**Keywords:** perovskite; laser annealing; crystallization; Raman

## 1. Introduction

Organic–inorganic halide perovskite solar cells [1–3] are among the promising photovoltaic materials due to their excellent power conversion efficiency (PCE), which reaches more than 20% [4–6], with a very low material cost. Now, they are widely used in photovoltaic and optoelectronic applications due to their broadband absorption, high optical absorption coefficient, very low exciton binding energy, long carrier diffusion lengths, and efficient charge collection [7]. Solution-processed organic–inorganic perovskite materials [8] with PCEs approaching those of crystalline Si solar cells gave rise to low-cost and highly efficient perovskite solar cells [9].

The device performances were determined by the crystallinity of perovskite films [10–12]. Therefore, the vital factor for a reliable perovskite device is the effective processing of the high crystalline perovskite structure with uniform film morphology. With the current technological trend toward flexible and wearable devices, the perovskite-based optoelectronic devices were among the good candidates. Simple and stable crystallization methods for the large-area flexible perovskite-based substrate are crucial.

Laser annealing techniques [13] can perform precise and robust crystallization for perovskite films. Chen et al. [2] showed the annealing effects of the CH₃NH₃PbI₃ perovskite film on solar cell applications with different thermal annealing temperature processes,

and indicated that the optimal thermal annealing temperature was 100 °C. Li et al. [14] also successfully introduced the preparation of homogeneous, dense-grained $CH_3NH_3PbI_3$ films with laser irradiation as a rapid crystallization approach. Jeon et al. [15] presented the laser crystallization of hybrid perovskite solar cells using a near-infrared (NIR) laser ($\lambda$ = 1064 nm). The laser annealing of $CH_3NH_3PbI_3$ perovskite film in a solar cell structure using pulsed Nd:YAG laser radiation with a wavelength of 1064 nm was shown by Malyukov et al. [16] and Trinh et al. [17]. This operation wavelength was located far from the absorption spectra of typical perovskite films. Direct laser annealing allows for the increase of photoluminescence over 130% and increased absorbance of over 300% in the near-IR range by Tiguntseva et al. [18].

Lots of excellent works have been carried out. However, the offline measurements by solar simulators [19] and a UV-Vis spectrophotometer [3,20] were performed to confirm the optoelectrical performances after the laser annealing processes. The optimized conditions were determined by empirical data, because too many variables were involved in the determination of meaningful predicted parameters, such as the laser operation modes, the laser wavelength, the sample substrates, and others. Quarti et al. reported theoretical and experimental Raman vibration analysis of $CH_3NH_3PbI_3$ perovskite films [21], and provided the structured information of the 2D and 3D of halide perovskite [22]. Polarised Raman signals can be confirmed as a non-destructive technique to characterize the crystallization degree of the perovskite films and microwires.

In this work, we prepared the perovskite microwires and performed laser-induced thermal annealing for perovskite microwires crystallization. This proposed optical system included the violet annealing laser and a micro Raman spectroscopy system. The crystallization of the pre-set area of the perovskite microwires can be performed by the violet annealing laser, and then checked by micro Raman spectroscopy. We showed how to optimize the laser annealing parameters and found the laser annealing conditions for the different substrates.

## 2. Materials and Methods

### 2.1. The Preparation of Perovskite Wires

Zhu et al. [23] presented how to convert the perovskite thin films into nanowires, and showed the potential of the flexible photodetector. Fabrication of self-assembly polycrystalline perovskite microwires (MWs) was also designed and performed by Zhu et al. [24]. In this work, the $CH_3NH_3PbI_3$ MWs were fabricated based on a fluid-guided, anti-solvent, vapor-assisted crystallization (FGAVC) approach. The 0.02 M of the $CH_3NH_3PbI_3$ precursor solution was prepared by dissolving $CH_3NH_3I$ and $PbI_2$ (molar ratio of 1.5:1) in $\gamma$-butyrolactone (GBL):DMF (50% and 0% $v/v$) binary solvents. Clean glass substrates were dipped with the $CH_3NH_3PbI_3$ precursor solutions for a few seconds, and then placed at a tilt angle of 5°–10° in a sealed glass bottle containing 1.5 ml of dichloromethane (DCM). After 6 h, 1 mL of DCM was dripped onto the substrate. Then, the substrate was removed from the bottles to wash out the remaining DMF, and the produced $CH_3NH_3PbI_3$ samples were baked at 100 °C for 30 min. With the FGAVC method, the MWs significantly became longer, and a gap existed between each one without aggregation. A large number of DCMs diluted the solution on the glass sheet. Thus, the MWs grew less in contact with each other. The image of CH3NH3PbI3 perovskite MWs using optical microscopy (OM) under a 10X objective lens is shown in Figure 1a, and the OM image of three separated MWs is shown in Figure 1b.

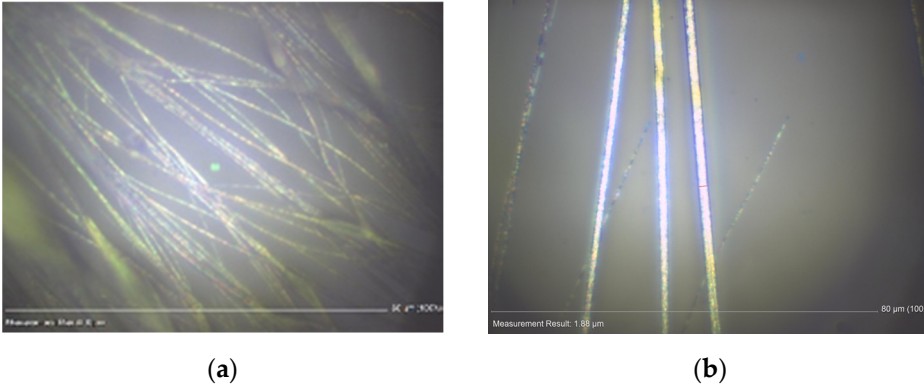

(**a**)                                          (**b**)

**Figure 1.** (**a**) The OM image of CH₃NH₃PbI₃ perovskite MWs using a 10X objective lens, and (**b**) the OM image of a single CH₃NH₃PbI₃ perovskite MW.

### 2.2. The Laser Annealing and the Raman Measurements

The simple optical layout is shown in Figure 2. A 405 nm violet diode laser (Lambda Beam, rgb photonics) with 36 mW output power and near-TEM00 spatial mode was used as the annealing laser. The electronic shutter after the violet laser was installed to control the exposure time. The wheeled neutral density (ND) filter was used to control the laser power. Process parameters of the laser annealing technique can be optimized for local intensities and exposure time. A laser beam can deliver the power-stable, near-Gaussian profile intensity distributions in the spatial domain. Here, the laser beam was scanned on a predetermined rectangular area with a 2D precision linear motor-driven stage. The laser intensity was clearly defined.

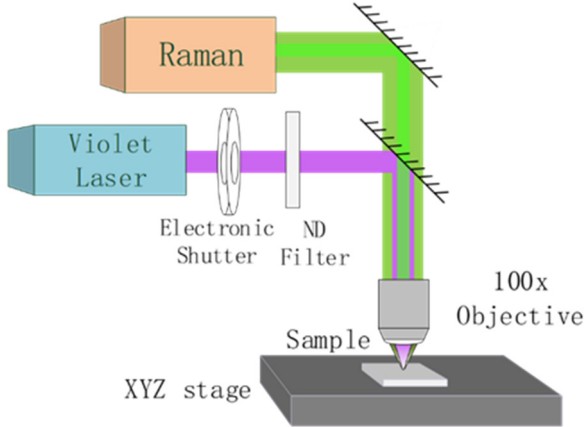

**Figure 2.** The optical scheme of laser-induced thermal annealing and the polarized micro Raman system. ND Filter: neutral density filter.

This micro Raman spectroscopy system (MRID, Protrustech Taiwan) was based on an optical microscopy (BX38, Olympus). The excitation laser, a narrow line width, linear-polarized, 532 nm diode-pumped solid-state laser, was installed with a polarized Raman spectroscopy system. The excitation laser power was kept as 0.5 mW to avoid damaging the perovskite MWs. This proposed in situ laser annealing and perovskite characterization for perovskite crystallization can perform the single-point crystallization and large-area crystallization with a 2D beam-scanning platform.

## 3. Results and Discussions

### 3.1. Laser-Induced Thermal Annealing Process

To find out the parameters of the annealing laser, we tried to focus the laser beam by using a 100X objective lens on the perovskite MWs on a glass slide. Here, the laser beam diameter was measured as 2.6 μm. The laser power was measured as 16 mW on the sample plate when a 36 mW laser beam was guided by the optical system and focused on the perovskite MWs. Figure 3a,b are the images of perovskite MWs before and after the violet laser beam illumination, respectively. The specific point on perovskite MWs was burned out when it was exposed to 6 kW/mm$^2$ laser intensity on 10 s time duration. The thermal image shown in Figure 3c was due to laser beam illumination on perovskite MWs; the image was taken using a thermal infrared camera. The thermal infrared camera was also used to monitor the laser annealing process to avoid laser damaging. The damage threshold was estimated as 53 kJ/mm$^2$, and the corresponding temperature was 110 °C, when the perovskite microwires were coated on the glass slide. The heating curve with time is shown in Figure 3d; it slightly differed due to the substrates. The heating curve on the glass was higher than on the polyethylene terephthalate (PET) film.

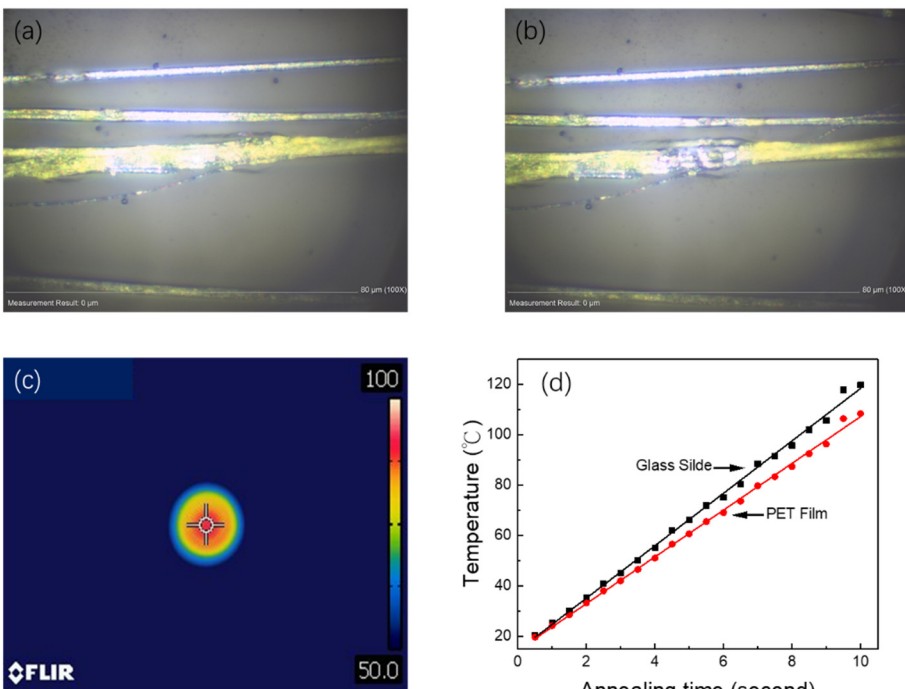

**Figure 3.** The OM image of a perovskite film (**a**) before and (**b**) after the laser beam illumination. A violet laser beam was focused by a 100X objective lens on the perovskite film. The laser beam diameter was measured as 2.6 μm. (**c**) The thermal image pictured by a thermal infrared camera due to laser beam illumination.(**d**) The heating curve with time.

### 3.2. The Crystallization Improvement of a Single-Point Laser-Induced Annealing

Figure 4a showed the OM image of a laser-induced annealing point on a perovskite MW. The Raman spectra before and after the laser-induced annealing are shown in Figure 4b. For pristine methylammonium lead iodide (MAPbI$_3$) MW, the Raman signals revealed the strongest band at ca. 94 cm$^{-1}$ and two bands at 74 and 105 cm$^{-1}$. After the laser-induced annealing process, however, the Raman spectrum was found to change significantly. We observed two distinct main bands at 74 and 105 cm$^{-1}$, and a considerable reduction of the intensity of the 94 cm$^{-1}$ band.

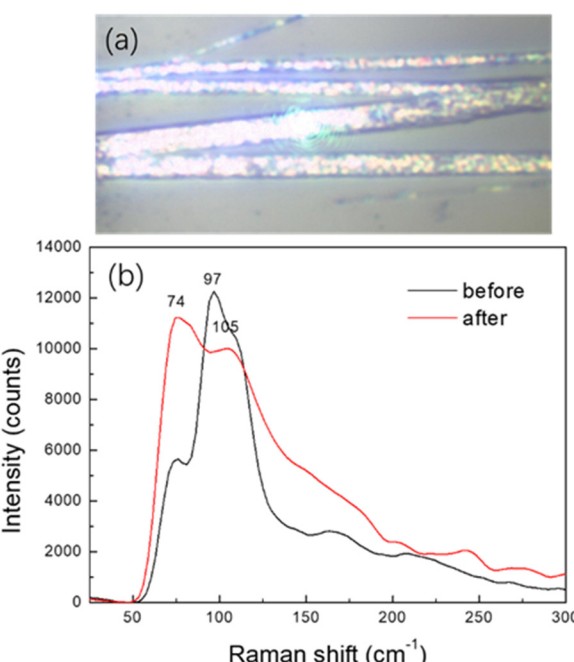

**Figure 4.** (**a**) The OM image of a laser-induced annealing point on a perovskite MW; (**b**) the typical Raman spectra before and after the laser annealing. Here, the violet laser power was kept as 17 mW, and the exposure time was 5 s.

The MAPBI$_3$ perovskite showed various bands between 40–300 cm$^{-1}$, associated with the librations of the MA organic cations and the deformation of the inorganic cage. Quarti et al. reported a theoretical and experimental Raman vibration analysis of CH$_3$NH$_3$PbI$_3$ perovskite films [21], and provided the structured information of the 2D and 3D of halide perovskite [22]. The Raman band position may change in the order of several cm$^{-1}$ for the different crystalline samples and excitation sources. The band at around 105 cm$^{-1}$ could be interpreted as the motion of the organic cations within the inorganic cage, and is a marker of the formation of nondegraded MAPBI$_2$ perovskite crystal. On the contrary, the band at ca. 97 cm$^{-1}$ was an indicator of the degradation product of PBI$_2$. Ledinsky et al. observed an increased intensity of the 94 cm$^{-1}$ band of the degraded MAPBI$_2$ perovskite thin films [25]. Therefore, the characteristic peaks at ca. 97 and 105 cm$^{-1}$ could be used to evaluate the crystallization quality of a single perovskite MW. Additionally, the normal modes below 90 cm$^{-1}$ were from various Pb-I stretching and bending vibrations of the inorganic cage. Therefore, the increased intensity of the band at 74 cm$^{-1}$ also suggested an improved crystalline quality of MAPBI$_3$ MW after the laser-induced annealing process.

The intensity ratio of a 105 cm$^{-1}$ peak over a 97 cm$^{-1}$ peak ($I_{105}/I_{97}$) was adopted for the crystallization quality coefficient. The violet laser power was kept as 17 mW, and the exposure time was 5 s. The $I_{105}/I_{97}$ ratio was picked, and was calculated as 0.81 before the laser-induced annealing, and as 1.07 afterward.

### 3.3. The Crystallization Improvement of the Perovskite MWs Due to the Laser Annealing Process

The laser beam was scanned on a programmable, predetermined rectangular area with a 2D precision linear motor-driven stage. The scanning range had a 30 μm × 30 μm square area with a 0.5 μm step, and the scanning speed was 0.5 μm/s. The laser beam was scanned with a prearranged period. Typically, the laser power can be tuned for the optimized annealing condition. The Raman signals were used for identifying the crystallization of the perovskite MWs after the laser annealing processes. Then, the best annealing conditions could be found out.

In Figure 5a, we showed the OM images of perovskite MWs before the laser annealing process. Here, the laser beam diameter was measured as 2.6 μm, and the scanning area

was 900 $\mu m^2$. The Raman mapping of the crystallization improving factor ($I_{105}/I_{97}$ ratio) before the laser annealing process is shown in Figure 5b. After the laser annealing process, the sample surface was cleaned by laser beam sweeping, and its image is shown in Figure 5c. The crystallization of the perovskite MWs was increased by the laser-induced thermal annealing according to the Raman mapping of the crystallization improving factor ($I_{105}/I_{97}$ ratio), as shown in Figure 5d. We can conclude that this approach was useful for the crystallization of the perovskite MWs.

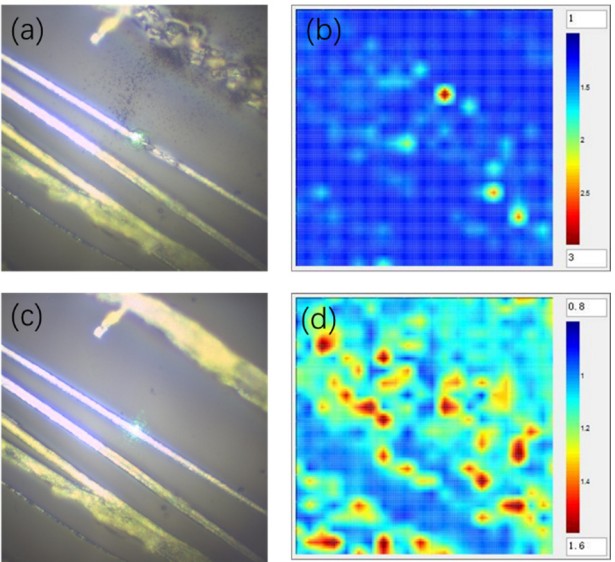

**Figure 5.** (**a**) The OM images of perovskite MWs; (**b**) the Raman mapping of the crystallization improving factor (I105/I97 ratio) before the laser annealing process; (**c**) the OM images of perovskite MWs; and (**d**) the Raman mapping of the crystallization improving factor ($I_{105}/I_{97}$ ratio) after the laser annealing process.

## 4. Conclusions

We introduced the in situ optical scheme of laser-induced thermal annealing and the characterization for perovskite MW crystallization. This proposed optical system provided a real-time laser annealing optimization process with crystallization quality monitoring that can be well-operated. This design can be extended to other laser fabrications. Compared with the NIR lasers [14,17] proposed by other groups, a 405 nm violet laser located at the absorption peak of perovskite MWs can provide reliable controllability of perovskite crystallization with tens of mW of laser beam. Rapid, high-performance, laser-induced thermal annealing can be realized on a PET flexible polymer and on hard glass substrates. The heating process due to the different substrates can be estimated by an online Raman measurement. Furthermore, elliptical laser beams were introduced with online Raman spectroscopy that can be applied to large-area laser processing. For solution-based, large-area production of flexible/wearable perovskite electronics and optoelectronics, or the relevant organic−inorganic hybrid materials, this laser processing offers a platform for testing and manufacturing. This proposed approach was useful for the crystallization of the perovskite MWs. However, it still cannot provide information about how much the improvement of the crystallization has occurred. The work for figuring out the crystallization degree is still ongoing.

**Author Contributions:** Conceptualization, X.C. and H.-L.C.; Raman measurement, Z.W.; Sample Preparation, R.-J.W.; Data Analysis, H.-L.C.; writing and supervision, H.-C.C. All authors have read and agreed to the published version of the manuscript.

**Funding:** This research was funded by the Ministry of Science and Technology (MOST 109-2221-E-006-149) and by the Fundamental Research Funds for the Central Universities, China (DUT18RC(3)047 and DUT20RC(5)028)

**Data Availability Statement:** The data presented in this study are available in the article.

**Conflicts of Interest:** The authors declare no conflict of interest.

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
