# Peer review of "Laser-Induced Thermal Annealing of CH3NH3PbI3 Perovskite Microwires"

_photonics, doi:10.3390/photonics8020030_

Round 1

Reviewer 1 Report

The subject is interesting. The authors propose a more efficient route to produce crystalline perovskite microwires. However is advisable to improve the discussion.

In the Materials and Methods section (Page 2), the authors have been mentioned two ways to produce perovskite wires, one about nanowires and the other about microwires. Are these methods related to the method chosen by the authors to produce the CH3NH3PbI3? The efficiency of perovskite materials is related to their processing routes. What were the advantages that led the authors to choose the method shown in the present paper? Explain it in the text and reference the method.

(Page 3) Figure 3b shows a point of perovskite burned by the laser. What is the consequence of burning the material? How it can be avoided?  

(Page 4) The first time that MAPBI3 appears during the text is in line 148. Define its meaning.

(Page 5) It is known that Raman spectra are related to the short-range ordering of materials structures and their respective vibrational modes. However, the authors did not associate the bands with the vibrational modes that gave rise to them. Regarding the text, the band at 97 cm-1 is an indicator of the degradation product of PBI2. After the laser annealing process is possible to note that the band located at 97 cm-1 vanished. By the results is possible to affirm that the laser avoids the degradation of materials? This behavior is important for the applicability of the material? Also, the authors mentioned that the intensity ratio of 105cm-1 peak over a 97 cm-1 peak was adopted for the crystalization quality coefficient. Related it with the degradation of PBI2 and the perovskite structural ordering.

(Page 6) In the Conclusion section, the authors say that the crystallization due to laser can enhance the photoelectric conversion efficiency, increase the carrier mobility, and reduce the recombination rate inside. What were the analyzes performed by the authors that support this statement?

Reviewer 2 Report

In the manuscript, authors employed a 405-nm CW laser to induce the crystallinity of organic-inorganic perovskite microwires, and successfully demonstrated crystallization of the perovskite microwires. There are a few questions that need to be addressed before considering for publication.

  1. In Figure 3, the spot on the perovskite MW, which was illuminated by the violet laser looks damaged. Did that happen due to an abrupt crystal growth at the spot or a real damage from the laser illumination?
  2. In Figure 3c, the Chinese characters need to be translated into English for readers.
  3. GIWAXs or a simple XRD might help to confirm the crystal growth and orientation.
  4. Grammar needs to be double checked. For instance, in this sentence, “We showed how to optimize the laser annealing parameters and also pointed out that laser annealing conditions differed with the substrates.”, the meaning of “laser annealing conditions differed with the substrates” is vague. Please check the grammar in the manuscript and make clear the sentences.

Reviewer 3 Report

The reported observations are interesting and worth to be reported. However, I have some issues to be solved before the publication of the manuscript.

  • The author mentioned that the irradiated laser had 2.6 um of beam diameter and the power was set as 16 mW. However, it is hard to notice where is the irradiation position. The samples were scanned or the laser exposed to only one position. The author should describe how much area was under the effect of laser annealing. In figure5, the guide for the eye needs to find where is the irradiated position and scale bar also should be included.

  • The authors insist that the improvement of the crystallization and the Raman shift were analyzed as evidence of improving crystallization. The Raman data can tell about the change of phase before and after irradiation, due to induced heat from the laser, however, it is hard to provide the information on how much the improvement of the crystallization has occurred. I corned that “improving” can lead readers to misunderstand.

  • The authors set the laser power at 16mW. Do you have any reason for it? In other words, do you have an optimization process to determine laser power and time? If it performed, please add the description about them.

  • Typos should be corrected. For example, “6kW/mm2”.
